# Tailoring the Emission Wavelength of Color Centers in Hexagonal Boron Nitride for Quantum Applications

**DOI:** 10.3390/nano12142427

**Published:** 2022-07-15

**Authors:** Chanaprom Cholsuk, Sujin Suwanna, Tobias Vogl

**Affiliations:** 1Abbe Center of Photonics, Institute of Applied Physics, Friedrich Schiller University Jena, 07745 Jena, Germany; 2Optical and Quantum Physics Laboratory, Department of Physics, Mahidol University, Bangkok 10400, Thailand; sujin.suw@mahidol.ac.th; 3Fraunhofer-Institute for Applied Optics and Precision Engineering IOF, 07745 Jena, Germany; 4Cavendish Laboratory, University of Cambridge, Cambridge CB3 0HE, UK

**Keywords:** single photons, defects, quantum emitters, density functional theory, strain tuning, charge states

## Abstract

Optical quantum technologies promise to revolutionize today’s information processing and sensors. Crucial to many quantum applications are efficient sources of pure single photons. For a quantum emitter to be used in such application, or for different quantum systems to be coupled to each other, the optical emission wavelength of the quantum emitter needs to be tailored. Here, we use density functional theory to calculate and manipulate the transition energy of fluorescent defects in the two-dimensional material hexagonal boron nitride. Our calculations feature the HSE06 functional which allows us to accurately predict the electronic band structures of 267 different defects. Moreover, using strain-tuning we can tailor the optical transition energy of suitable quantum emitters to match precisely that of quantum technology applications. We therefore not only provide a guide to make emitters for a specific application, but also have a promising pathway of tailoring quantum emitters that can couple to other solid-state qubit systems such as color centers in diamond.

## 1. Introduction

Optical quantum technologies promise groundbreaking applications with quantum communication [1], quantum computing [2], and quantum metrology [3]. Using single photons as flying qubits for quantum information processing has several advantages: photons travel at the speed of light, resulting in a fast transmission in communication systems, and they only interact weakly with the environment, which translates into robust qubit states [4,5,6]. Moreover, single quantum states can be easily manipulated with linear optics to process quantum information. These advantages, however, come at the expense of difficult multi-qubit operations, as the photons do not interact directly with each other. The controlled NOT (CNOT) gate required for universal quantum computing is thus not realizable with linear optics [7] and needs a nonlinear medium as a mediator (e.g., through cross-phase modulation [8]). This necessity can be relaxed with one-way or measurement-based quantum computing, which uses entangled qubits as a resource and then manages to perform universal quantum computations with single qubit gates realized with linear optics [9,10]. The generation of entangled states from single photons, however, requires any single photon source (SPS) to be of high quality.

Many well-known single photon emitting systems have been investigated, including semiconductor quantum dots [11,12], color centers in solid-state crystals such as diamond [13], silicon carbide [14] or indium selenide [15], as well as trapped ions [16]. While single photons emitted from quantum dots have near-ideal photon purity and indistinguishability [17], they require low temperatures for operation. The widely investigated NV${}^{-}$ center in diamond suffers from a low Debye–Waller (DW) factor below 4% at room temperature [18] which limits its optical coherence due to the strong phonon sideband. Other color centers in diamond such as SiV${}^{-}$ have a much higher DW factor of up to 70%, but only a relatively low quantum efficiency of 3.5% [19]. The SiV${}^{-}$ centers in SiC host crystals also have high quantum efficiencies around 70% but medium DW factors around 33% [14]. This fuels the search for novel quantum emitters with both high quantum efficiencies and DW factors at room temperature.

The recently discovered color centers hosted by two-dimensional (2D) hexagonal boron nitride (hBN) [20] have demonstrated DW factors as high as 82.4% [21] and quantum efficiencies of 87% at room temperature [22], alongside bright and pure single photon emission. Due to the 2D geometry of the host crystals, it is easily possible to integrate the emitters with waveguides and fiber networks [23,24]. In addition, the robustness of the emitters, their long-term stability, and a fast radiative decay lifetime allowing high repetition rates [21,25], has led to the general understanding that these emitters can be used in practical quantum information processing applications. In fact, single photons emitted from hBN have been used for quantum random number generation [26,27] and single photon interferometry [28]. Their linewidth at room temperature, however, still needs substantial improvement in order to be used for optical quantum computing [29].

Around the nature of the emitters, there has been a large debate among the community, due to the large distribution of zero-phonon lines (ZPLs) [30,31]. Recent works have correlated experimentally observed emission spectra with theoretically calculated defect complexes [32]. For instance, the emitter with a zero-phonon line around 2 eV was assigned to the (2)${}^{3}$B${}_{1}$ to (1)${}^{3}$B${}_{1}$ transition of the V${}_{N}$C${}_{B}$ defect [33]. Likewise, the C${}_{B}$C${}_{N}$ complex was shown to be responsible for the 4 eV emission via calculations of the coupling to the vibrational degrees of freedom [34] and via delta self-consisted field (ΔSCF) calculations [35,36]. The influence of carbon in the emitter formation was also shown experimentally [37]. Besides carbon-based defects, oxygen complexes such as O${}_{B}$V${}_{N}$O${}_{B}$ [38] and the negatively charged boron vacancy V${}_{B}^{-}$ [39] are believed to be responsible for quantum emission in the visible spectrum. Most theoretical studies take the approach of matching the calculated spectra to experimentally observed data, e.g., the photoluminescence spectrum [31,32,38,40,41,42].

In this work, we calculate a large number of defect complexes in order to uncover their radiative transitions using spin-polarized density functional theory (DFT) with the Heyd–Scuseria–Ernzerhof (HSE06) functional. In contrast to previous works, we do not aim to match computational results with experimental data, but rather to discover new defects which have radiative transitions at important wavelengths for quantum technologies. This should facilitate one to fabricate defects with the desired wavelength for specific applications. Frequency conversion schemes with non-ideal efficiency are thus not necessary. Our comprehensive study includes the electronic structure of 267 defect complexes with substitutional atoms from periodic main groups III through VI, transition metals, and multi-defect complexes. For a selected subset of defects, we have also investigated the effects of their charge states (neutral and ±1). Ultimately, we develop criteria to identify promising defects based on the transition type (radiative/non-radiative), localization (deep/shallow), and transition energy. Based on these criteria, we select emitters whose wavelengths are close to the important ones for quantum technologies and apply strain-tuning to tailor their emission wavelengths. As exemplary wavelengths, we chose here the resonance wavelengths of solid-state quantum emitters and qubits in diamond and silicon carbide, typical quantum memory wavelengths in rare-earth ion doped crystals and alkali vapors, as well as low-loss telecom wavelengths for long-distance quantum communication. We therefore provide a comprehensive list of promising quantum emitters in hBN and a mechanism to tailor them to enable potential applications in quantum technologies.

## 2. Computational Details

### 2.1. Density Functional Theory

All spin-polarized DFT calculations have been performed using QuantumATK (version S-2021.06) [43], which utilizes numerical linear combination of atomic orbitals (LCAO) double ζ polarized basis sets [38]. The used pseudopotentials and basis sets (which influence the accuracy of our DFT calculations) are internally constructed by our DFT calculator from a frequently benchmarked and reviewed database for all elements to provide the most accurate results. Point-like defects were created in the center of a 7×7×1 supercell (see Figure 1), which was proven to be sufficiently large to exclude any defect–defect interaction of neighboring cells. For the lattice structural optimization, only internal coordinates were allowed to relax using a 5×5×1 Monkhorst–Pack reciprocal space grid [44] until all forces were below 0.01 eV·Å${}^{-1}$ and the total energy convergence reached $10^{-4}$ eV. It is noteworthy that all supercell calculations were initially optimized from the relaxed unit cell to save computational time. A vacuum layer of 15 Å was added to minimize the van der Waals interaction between layers. Since this work mainly focuses on the electronic band structure and optical transitions, we employed the HSE06 functional [45] to avoid underestimating the band gap. We note that even using the HSE06 functional, DFT has only a finite accuracy. The predicted band structures are nevertheless quantitatively correct and agree well with experiments, as a recent study comparing different DFT functionals to calculate quantum emitters in hBN has pointed out [41,46]. Moreover, HSE06 has demonstrated very good agreement with experiments (and in particular better than functionals in the generalized gradient approximation) [46]. To capture all possible states, a dense 11×11×1
*k*-point centered at the Γ point was implemented.

### 2.2. Strain-Tuning

In order to being able to manipulate the transition energy, bi-axial strain was applied to the crystal structures by elongating and compressing them in both *a* and *b* directions isotropically (for definition of the axial directions see Figure 1). To create strain in the crystal lattice, the built-in pressure function in QuantumATK has been used. The pressure is applied by a 3×3 stress tensor, where we define identical values for the xx and yy components (and all other components set to zero) to simulate bi-axial strain. The geometry is relaxed again and the applied strain *s* can be calculated using the initial length of the unstrained lattice parameter $L_{0}$ and the length difference of the strained lattice parameter ΔL, where the strain is given by $s=\frac{\Delta L}{L_{0}}$.

### 2.3. Criteria for Promising High Quality Defects

In principle, the photoluminescence properties of any quantum emitter depend on the Huang–Rhys factor, Debye–Waller factor, its quantum efficiency (QE), and the excited-state lifetime. These excited-state properties depend on the electronic transition, which can be classified in general into two types: radiative and non-radiative transitions. Both transition types are disparate in the sense of the electron-phonon coupling. While the radiative transition can happen without coupling to any phonon (i.e., into the zero-phonon line), the non-radiative transitions dissipate the excitation mostly through phonons. Ideally, any quantum emitter should feature a low Huang–Rhys factor, a high DW factor, high QE, and short excited-state lifetime.

To select suitable defects for quantum technologies among the large number of investigated complexes, we define the following criteria for promising quantum emitters:Electronic transition type: the defect should form a two-level system, where the transition between the highest occupied and the lowest unoccupied defect state is radiative. We determine this by considering the imaginary part of the dielectric function or the optical absorption spectrum. Non-radiative transitions do not exhibit any characteristic peaks;Transition energy: the emitter should have an optical emission wavelength useful for quantum technology applications. We estimate this from the energy difference between the ground (highest occupied) and excited (lowest unoccupied) states, which correspond to the characteristic peak observed in the imaginary part of dielectric function;Localization in the band structure: defect states can naturally occur in either shallow or deep regions in the band gap. For a high quantum efficiency at room temperature, the states should be well isolated from the band edges or other defect states (i.e., deep defects). Those deep-lying states should also exhibit a flat line in both the density of states and the electronic band structure, as this implies they have the same energy in every high symmetry point. In other words, such deep-level defects are well isolated from interaction with neighboring atoms.

It is important to point out that this work considers only first-order transitions between the highest occupied state and the lowest unoccupied state. While there might be optically allowed transitions to higher unoccupied states possible, such excitation would have multiple decay channels which reduces the quantum efficiency of any given transition. Moreover, based on the first-order electronic transition, half of all defects are classified as being non-radiative. In another ongoing work we employ the ΔSCF method to uncover the allowed optical transition pathways of those defects from the zero-phonon line.

## 3. Results and Discussion

### 3.1. Transition Energies

We first benchmark the electronic structure of a pristine hBN monolayer with other reports in the literature. The lattice parameters, band gap, and the atomic orbital contribution in the electronic band structure are well consistent. To be more precise, our calculation with the HSE06 functional yields a direct electronic band gap of $E_{g}=5.99$ eV (see Appendix A), which agrees well with experimental results [47] and that of calculations in the $G_{0}W_{0}$ quasiparticle approximation [48] reporting values around 6 eV.

Then, the electronic band structures of different defect complexes were calculated and classified according to our criteria mentioned above. Figure 2 shows the distribution of transition energies or emission wavelengths of 92 identified radiative defects. The histogram in the inset shows that most fluorescent defects hosted by hBN have transition energies in the range of 1.6 to 2.6 eV, while a large majority is not optically active. Notably, many transitions below 0.75 eV and above 3.25 eV mostly exhibit radiative transitions or transitions between degenerate states, which can also be treated as radiative. It becomes clear that the transition energy does not correlate with the position of any impurity or dopant within one of the periodic main groups. To be more precise, although the defects may possess the same number of valence electrons owing to the same periodic main group, the overlapping between the specific element and the host hBN can be distinct and independent, conditional on how the spin polarization of the electrons is minimized. However, we observed that impurities from group III though VI contribute to transition energies typically ranging between 1 and 4 eV, while transition energies below 1.5 eV originate mostly from transition metals as dopants (the O${}_{N}$S${}_{N}$ is the exception).

#### 3.1.1. Group III Dopants

We now turn to a more in-depth analysis of the defect complexes, starting with impurities from group III, namely Al, In, and Ga substituted into the structure at boron or nitrogen lattice sites. Figure 3 (see also Appendix A) shows that any dopant (also outside group III) retains the large band gap value, despite shifting the Fermi energy. This implies that intrinsic room temperature quantum emission in general is possible, independently of the dopant.

The Al${}_{B}$ and Ga${}_{B}$ centers do not induce any states between valence and conduction band (VB and CB, respectively), because their equal number of valence electrons compared to boron. In other words, there is no additional free charge carrier left to occupy any additional states. For the In${}_{B}$ center, however, a new deep (but unoccupied) state appeared. For Al-based defects, only the Al${}_{B}$V${}_{B}$, Al${}_{N}$, and Al${}_{N}$V${}_{B}$ defects have radiative transitions around 2 eV, whereas for In-based defects, possible transitions were found for the In${}_{B}$V${}_{B}$, In${}_{B}$V${}_{N}$, and In${}_{N}$V${}_{B}$ center. In the case of Ga-based defects, we did not find any first-order transitions.

#### 3.1.2. Group IV Dopants

Out of group IV we selected C, Si, Ge, and Sn for potential defect complexes (see Appendix A). For C-based defects, we found a rich variety of optically active defects, namely the C${}_{B}$V${}_{B}$, C${}_{N}$V${}_{B}$, C${}_{N}$V${}_{N}$, C${}_{B}$C${}_{N}$, C${}_{B}$C${}_{N}$V${}_{B}$, and C${}_{B}$C${}_{N}$V${}_{N}$ centers. This is consistent with previous reports of C-based quantum emitters [35,36,37,49]. The C${}_{B}$ defect for example inherits one occupied and one unoccupied defect states, confirming that the structure has one additional free electron from carbon. Similarly, C${}_{N}$ also has one more free electron to create one unoccupied defect state. The electronic structures of both defects correspond to one spin-up occupied defect state and one spin-down unoccupied defect state as reported by previous spin-polarized DFT calculations [49,50,51]. Interestingly, bi-carbon defects such as C${}_{B}$C${}_{N}$ exhibit degenerate defect states, also consistent with previous work [36]. Our transition energies at 2 and 4 eV agree well with other zero phonon line calculations [35,36,49], validating our methodology. In contrast, for Si-, Ge-, and Sn-based defects, we found first order transitions of deep-level defects with energies in the range of 1.8 to 2.8 eV.

#### 3.1.3. Group V dopants

Out of group V we selected P, As, and Sb for potential defect complexes (see Appendix A). Notably, doping with phosphorus yields only in the case of the P${}_{B}$V${}_{B}$ center an optical transition at 2.56 eV. Doping with arsenic in turn induces a large variety of transition energies at 5.24 eV (As${}_{N}$), 1.72 eV (As${}_{N}$V${}_{B}$), and 3.16 eV (As${}_{N}$N${}_{B}$V${}_{N}$). For the Sb-based defects studied, the Sb${}_{B}$V${}_{B}$ center is the only interesting defect for quantum emitters. Overall, this result suggests that the first-order transitions in group V-based defects are mostly non-radiative. We believe this is due to the structures having interacting unpaired electrons that can minimize the energy. This flips the spin such that the systems take a state of lowest total energy.

#### 3.1.4. Group VI Dopants

For group-VI dopants (see Appendix A) we found again a large variety of allowed transition energies, in particular for oxygen-based complexes (except for O${}_{N}$V${}_{N}$), consistent with experimental work [21,52]. We note that the O${}_{N}$ and O${}_{N}$V${}_{B}$ defect states are near the band edges (i.e., shallow defects) and are therefore not meeting our criteria for suitable quantum emitters. We also note that previous DFT calculations have not reported any transition for the O${}_{N}$V${}_{B}$ and O${}_{B}$V${}_{N}$ defects [38], where we found some at 0.34 and 4.17 eV, respectively. This inconsistency can be explained by the different functional (HSE06) we used for our calculations. It was later pointed out that using non-hybrid functionals can underestimate the band gap [41,46]. For S- and Se-based we did not find any defects meeting all criteria for useful quantum emitters.

#### 3.1.5. Transition Metal Dopants

Transition metals being doped into the hBN lattice induce a a high density of localized states (see Appendix A), which is due to the high hybridization of transition metals. We found Er-based defects fit our selection criteria very well, as all of them yield the radiative transitions, ranging from 1.09 to 3.43 eV. In contrast, none of vanadium-based defects have a radiative transition. In the case of Ti- and Re-based defects, there are both radiative and non-radiative defects with a selection of different energy differences.

#### 3.1.6. Multi-Defect Complexes

We now expand our study to multi-defect complexes consisting of at least two impurities from different periodic groups (e.g., Al${}_{N}$Sb${}_{B}$, see Appendix A). We found that these defect complexes lead to a larger range of possible transition energies ranging from 0.58 to 5.60 eV, therefore covering almost the full band gap of hBN. From an experimental point of view, however, it remains a challenge to fabricate this type of defect, as different atomic species need not only to be implanted randomly but also at adjacent lattice sites.

### 3.2. Defects for Quantum Technology Applications

Our results so far show a broad coverage of the optical transition lines of hBN defects in the visible spectrum, with a few also in the near-infrared (for a complete list see Appendix A). The question arises, if they can be used for quantum technology applications. To demonstrate this, we selected a number of important wavelengths for quantum technology applications. To represent solid-state quantum emitters and qubits we chose color centers in diamond (with the NV and group IV centers [53]) and in silicon carbide (with the silicon vacancy [54]). Quantum memories are represented by rare-earth ion doped crystals (such as Pr${}^{3+}$:Y${}_{2}$SiO${}_{5}$ and Tm${}^{3+}$:Y${}_{2}$SiO${}_{5}$ [55]) and alkali vapor-based memories (e.g., sodium, rubidium, and caesium where the D1 or D2 transitions are commonly used [55]). For long-distance quantum communication, the telecom windows are crucial with the first window at 850 nm (now commonly used in free-space communications), and the telecom O- and C-bands at 1330 and 1550 nm, respectively. For almost all of these applications, we found a suitable hBN quantum emitter fulfilling our selection criteria and with a transition near the specific wavelength of the other quantum system, promising efficient coupling to it (see Table 1). Sometimes even multiple hBN defects are a possibility, or a defect is suitable for more than one application. We note that for some applications we only found transitions in the case of charged defects, which are discussed in Section 3.3.2. Moreover, in the case of the Sn${}_{B}$V${}_{B}$, Er${}_{B}$V${}_{B}^{+}$, and Er${}_{B}^{+}$ defects, the transitions are between the valence band or a degenerate (ground) state and an unoccupied (excited) state, which technically do not meet our criteria for deep defects. As the excited state in these cases is not degenerate, however, they can only be excited once as the Pauli exclusion principle forbids occupying the state with two electrons with the same spin direction. These defects therefore can, in principle, act as single photon emitters. Unfortunately, at 637 nm (ZPL of the NV${}^{-}$ center in diamond) we only found the Al${}_{N}$ defect which has a nearby transition between degenerate ground and excited states. This implies that this defect should rather emit the two photon Fock state, which can be useful in quantum interference experiments with many-particle states [56]. Experimentally, however, we have observed hBN quantum emitters with ZPLs around 640 nm [57], which implies that the experimentally observed emitters were probably not included in this theoretical study.

### 3.3. Tailoring the Transition Energy for Quantum Technology Applications

While we have identified hBN defects compatible with common quantum technology applications, their emission wavelength is only near the desired wavelength for the specific application. In the case of narrow transitions (e.g., the Rb-D2), this can drastically reduce the coupling efficiency to the other quantum system. Therefore, a fine tuning mechanism of the transition is required. This would make it possible to select an hBN emitter close to the target wavelength and apply this tuning method.

#### 3.3.1. Strain Tuning

Strain often breaks the symmetry and modifies the local structure of the atomic orbitals, thereby shifting the energy of the system [58]. This mechanism is particularly effective for 2D materials [33,59,60]. Strain-tuning of quantum emitters in hBN has been demonstrated experimentally shortly after their initial discovery [61]. We can apply this mechanism to tailor the transition energy with external bi-axial strain. We note that even using the HSE06 functional, DFT has only a finite accuracy. The exact required strain value would have to be determined experimentally. Moreover, due to typical residual stress in the crystal lattice, the ZPLs of hBN quantum emitters anyway spread around their zero-stress ZPL [57]. This also requires measuring the ZPL experimentally and *in-situ* tuning until the desired ZPL is achieved. Nevertheless, our results represent qualitatively the correct experimentally required strain.

In Figure 4 we present strain-tuning for a selection of identified compatible defects for quantum technology applications and tailor their emission to the precise wavelength of the corresponding quantum application (for the complete band structures see Appendix A). For example, for the S${}_{B}$V${}_{B}$ to be tuned into resonance with the PbV${}^{-}$ center in diamond at 552 nm, we need to apply 0.10% of strain. We found that predicting the required strain is infeasible, because the local structures with their defects react differently to the same amount of strain. Moreover, the strain can break the C${}_{2v}$ symmetry of the defects. Our results suggest the lattice deformation can be classified into in-plane (e.g., In${}_{B}$V${}_{N}$, Al${}_{N}$, and O${}_{N}$S${}_{N}$) and out-of-plane deformations (the remaining defects shown in Figure 4). As expected, the out-of-plane deformation results in larger wavelength shifts in the range of 50 to 250 nm, while for in-plane deformation this range is narrower with 30 to 40 nm. This is consistent with the fact that out-of-plane deformation considerably alters the symmetry of the local structures compared to in-plane.

As one might expect, many defects exhibit a linear relationship between strain and transition energy. Some, however, also show a nonlinear behavior (see Figure 4). This effect depends on the interplay between symmetry, initial strain and defect configuration such as location and size [62]. Generally speaking, we can express the strain Hamiltonian as $H=\sum_{ij}B_{ij}\epsilon_{ij},$ where $B_{ij}$ are the electronic operators, and $\epsilon_{ij}$ are the strain tensor components. When a quantum emitter is embedded in hBN, the system’s strain principal axes can change due to symmetry breaking of the defect. Depending on the direction and magnitude of the stretching or compressing, $\epsilon_{ij}$ can be a nonlinear function of the strain *s*. Therefore, the eigenvalues of *H* also nonlinearly depend on *s*. We note that even using the hybrid HSE06 functional, the DFT method still has a finite accuracy. It is therefore possible that, e.g., the trend of the S${}_{B}$V${}_{B}$ defect in Figure 4a is purely linear. The afore mentioned nonlinear effects e.g., on the In${}_{B}$V${}_{N}$ defect in Figure 4a, however, are larger than this accuracy, implying that the error of our DFT calculations is not aliasing as a nonlinear strain-wavelength relationship. Nonlinear strain dynamics have also been reported previously [58,61]. It was speculated that the initial conditions of the host crystal flakes with wrinkles, cracks and lattice mismatch from sample fabrication cause different local bond lengths and therefore an unpredictable Stokes shift.

It is worth noting that strain can also activate quantum emitters in hBN [63]. A model to qualitatively describe this is the donor–acceptor (DAP) model. Based on the DAP mechanism discussed in previous reports [49,64], there is an optimal bond length between donor and acceptor sites for strong photoluminescence. This suggests that when the local structure is under strain, the bond length between the active donor and acceptor is modified to the distance where the structure is optically active, but this is not necessarily where the structure is most stable (i.e., the zero-strain relaxed structure). This also implies that optically inactive defects that have been discounted due to our selection criteria could be turned active with strain. An in-depth study of this, however, is beyond the scope of this work.

#### 3.3.2. Influence of the Charge-State

So far, we have only considered neutral charge states. We finally also expand our study to charged defects (single positive/negative). We divide the defects in Appendix A into two groups based on whether they have a radiative or non-radiative first-order transition. Single-charging the defect (i.e., adding/removing one electron) naturally shifts the Fermi energy, as one more/less state is occupied (see Appendix A for the complete band structures). A positive charge can depopulate the top-most occupied state, while the negative charge can lower the bottom-most unoccupied (which is then occupied). In many cases, the transition type is conserved, meaning an optically active transition remains active under charging. We note that the level structure sometimes also completely changes. Ionization and recharging in general can lead to large shifts of the first-order transition, an effect that is known in diamond (e.g., comparing the ZPL of the NV${}^{0}$ center at 575 nm and the NV${}^{-}$ center at 637 nm [65]). Our findings are consistent with calculations of charged C-based defects [49].

While charging is therefore not a suitable in situ tuning mechanism, it is possible to generate totally new transition energies. It is worth noting that the well-studied V${}_{B}^{-}$ was not calculated in this study, but only the neutral boron vacancy. As we have mentioned, charging/ionization leads to new level structures, which explains why our results for the neutral boron vacancy differ from the literature with the negatively charged boron vacancy. Among our charge state calculations, we have identified compatible emitters for quantum technologies: the Er${}_{B}$V${}_{N}^{-}$ has a first-order transition near that of the V${}_{\mathrm{Si}}^{-}$ center in silicon carbide. The Er${}_{B}$V${}_{B}^{+}$ can couple to the Cs-D1 transition and the Er${}_{B}^{+}$ emits in the telecom C-band. This implies the latter can be used for long-distance quantum communication.

## 4. Conclusions

We have performed spin-polarized DFT calculations using the HSE06 functional that allowed us to characterize the electronic band structures of 267 defects hosted by hBN, as well as their transition type and energy. We have identified a large distribution of color centers with their transition energy peaking in a range from 0.25 to 5.50 eV. We have established criteria to select useful defects that can act as single photon emitters out of our new database. Moreover, we were able to match compatible hBN emitters for specific quantum technology applications, including quantum communication, quantum memories and the coupling to other solid-state qubit systems. Furthermore, we have theoretically demonstrated how the emission wavelengths of these hBN defects can be tailored to exactly match the specific wavelength of the quantum technology application. Our work therefore provides a guide for tailoring quantum emitters with a freely choosable target wavelength, as well as an important step towards the efficient coupling of different quantum systems in a large-scale quantum network.

## Figures and Tables

**Figure 1 nanomaterials-12-02427-f001:**
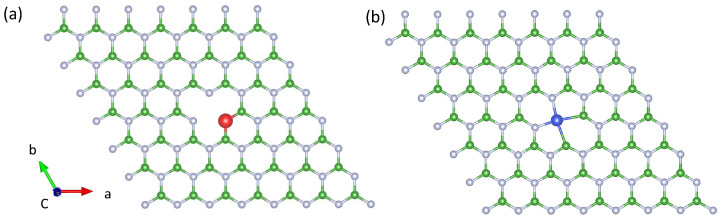
Examples of 7×7×1 supercells with defect complexes consisting of boron atoms (green), nitrogen (gray), aluminium (red), and titanium (blue). The *a* and *b* directions are in-plane. (**a**) Al${}_{N}$V${}_{B}$ defect, where a substitutional Al atom replaces a nitrogen atom next to a boron vacancy. (**b**) Ti${}_{\mathrm{BN}}$ where a Ti atom sits in the center of a bi-vacancy V${}_{\mathrm{BN}}$.

**Figure 2 nanomaterials-12-02427-f002:**
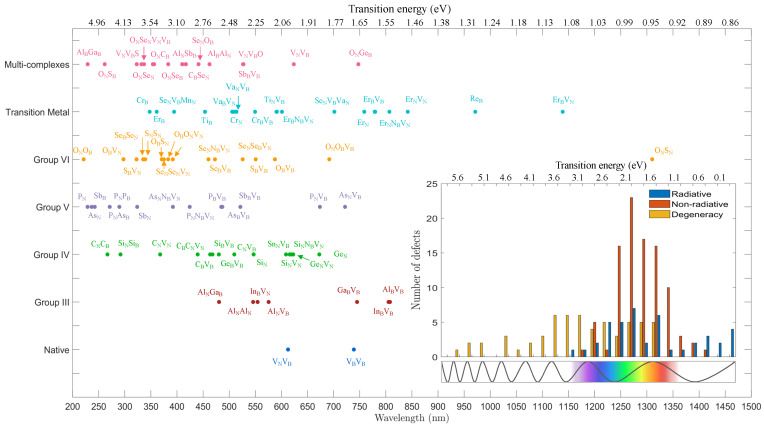
Distribution of optical transition energies (neutral charge states) of identified defect complexes in the optical spectrum. The defects have been classified according to the position in the periodic table of the involved impurities. The inset shows a histogram (with a bin width of 0.25 eV) of all 205 investigated (neutral) defects classified in their transition type (radiative, non-radiative, and degenerate). Note that in this article V denotes a vacancy, while for the chemical element vanadium the symbol Va is used.

**Figure 3 nanomaterials-12-02427-f003:**
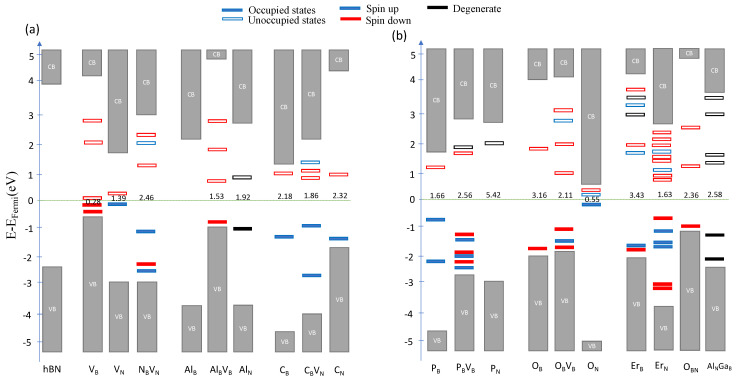
Modified electronic band structures identifying the defect states and the transition energy of some dopant representatives from (**a**) native defects and periodic groups III, IV, and (**b**) periodic groups V, VI, transition metals, and multi-defect complexes. The gray bar visualizes the states in valence and conduction bands, the filled (unfilled) bars represent the occupied (unoccupied) defect states, and the blue, red, black bars mean the defect states occupied by spin up, spin down and degenerate electrons, respectively. The transition energy between two-level systems is reported by the value near Fermi level. The O${}_{\mathrm{BN}}$ denotes an interstitial oxygen placed at the center of a bi-vacancy.

**Figure 4 nanomaterials-12-02427-f004:**
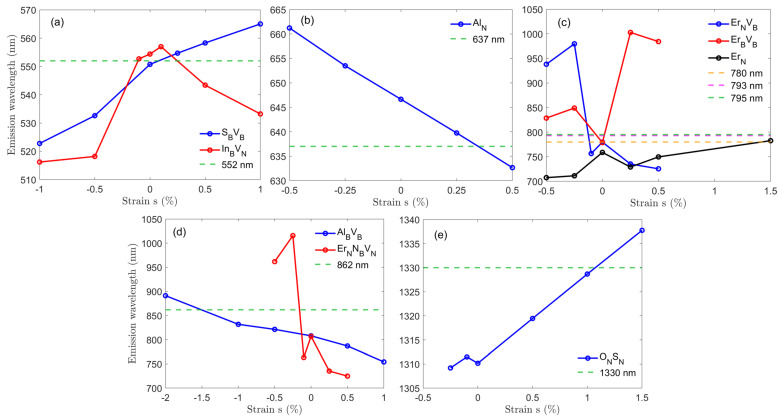
Emission wavelength as a function of bi-axial strain. The dashed lines mark the target wavelength and solid lines represent the emission wavelength of the defects at (**a**) 552 nm (PbV${}^{-}$ in diamond), (**b**) 637 nm (NV${}^{-}$ in diamond), (**c**) 780 nm (Rb-D2), 793 nm (Tm${}^{3+}$:Y${}_{2}$SiO${}_{5}$), and 795 nm (Rb-D1), (**d**) 862 nm (V${}_{\mathrm{Si}}^{-}$ in silicon carbide), and (**e**) 1330 nm (telecom O-band).

**Table 1 nanomaterials-12-02427-t001:** Suitable defects compatible with quantum technology applications.

Wavelength for Quantum Technology (nm)	Other Systems in Quantum Technology	Compatible hBN Defects	Defect Transition Energy (eV) / Wavelength (nm)
552	PbV${}^{-}$ (diamond)	S${}_{B}$V${}_{B}$	2.252/550.7
		In${}_{B}$V${}_{N}$	2.237/554.4
		In${}_{N}$V${}_{B}$	2.236/554.5
589	Na-D2	O${}_{B}$V${}_{B}$	2.110/587.6
590	Na-D1	Ti${}_{N}$V${}_{B}$	2.100/590.5
		V${}_{N}$V${}_{B}$Ti	2.097/591.3
602	GeV${}^{-}$ (diamond)	Er${}_{B}$N${}_{B}$V${}_{N}$	2.063/601.1
606	Pr${}^{3+}$:Y${}_{2}$SiO${}_{5}$	* Sn${}_{B}$V${}_{B}$	2.037/608.8
620	SnV${}^{-}$ (diamond)	V${}_{N}$V${}_{B}$	2.024/612.7
637	NV${}^{-}$ (diamond)	** Al${}_{N}$	1.918/646.6
738	SiV${}^{-}$ (diamond)	V${}_{B}$V${}_{B}$	1.678/738.8
780	Rb-D2	Er${}_{B}$V${}_{B}$	1.592/778.7
		Er${}_{N}$V${}_{B}$	1.590/779.6
793 795	Tm${}^{3+}$:Y${}_{2}$SiO${}_{5}$ Rb-D1	In${}_{B}$V${}_{B}$	1.540/805.2
		Er${}_{N}$N${}_{B}$V${}_{N}$	1.537/806.8
		Al${}_{B}$V${}_{B}$	1.535/807.9
850	Telecom-1	Er${}_{N}$V${}_{N}$	1.473/842.0
852	Cs-D2		
862	V${}_{\mathrm{Si}}^{-}$ (silicon carbide)	Er${}_{B}$V${}_{N}^{-}$	1.427/869.0
894	Cs-D1	* Er${}_{B}$V${}_{B}^{+}$	1.398/886.9
1330	Telecom O-band	O${}_{N}$S${}_{N}$	0.946/1310.2
1550	Telecom C-band	* Er${}_{B}^{+}$	0.789/1572.3

* Transition between VB or degenerate (ground) state and an unoccupied non-degenerate (excited) state. ** Transition between double-occupied (degenerate) ground and unoccupied degenerate excited state.

## Data Availability

The data set with the reported defects (sorted by type/wavelength and classified in terms of electronic transition type, transition energy, wavelength, and lattice deformation) can be found at https://doi.org/10.5281/zenodo.6826694 (accessed on 20 June 2022). The raw data is available from the authors upon reasonable request.

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
