# Peer review of "Tailoring the Emission Wavelength of Color Centers in Hexagonal Boron Nitride for Quantum Applications"

_nanomaterials, 2022, doi:10.3390/nano12142427_

Round 1

Reviewer 1 Report

The paper "Tailoring the emission wavelength of color centers in hexagonal boron nitride for quantum applications" by Tobias Vogl and co-workers describes a series of DFT computations on hexagonal boron-nitride 2D systems containing a large variety of dopants. The paper is well written and discussed and the conclusions arise convincingly from enough data. I have only a few concerns, described below, that should be considered before publication.

-I would like to see more details on the "double ζ polarized basis sets" employed. Authors are computing materials with heavy atoms such as various transition metals. Are the basis sets constructed in the same manner as light atoms? Were all electrons treated explicitly or the method used electron core-potentials? Are relativistic effects taken into account for these heavy atoms? If the treatment was different, please briefly discuss potential issues/limitations of the study coming from materials computed differently (regarding the basis sets) depending on the dopant.

-In the introduction, after the statement "we select emitters whose wavelengths are close to the important ones for quantum technologies" Please clarify which wavelengths you refer to.

Reviewer 2 Report

This is a very interesting and well presented paper making a convincing scientific case for specific colour centres in quantum technologies. In this context the paper is extremely topical and has the potential to make a big impact on the field, triggering more extensive experimental investigations and subsequent applications.

One extremely minor comment:
Should the figures in Fig 4 not have error bars and thus be smooth curves?

Reviewer 3 Report

This is a very nice work which will be highly appreciated both by experimentalists developing new generation of quantum emitters and theoreticians for benchmarking of dft methods. I think that the paper is well organized and should be published as is.

Probably, the only improvable point is the discussion of the choice of hse06. In my opinion, a bit more thorough comparison of this functional with literature data on the other ones used for band structure calculations would be helpful for a potential reader.
